# Computational model of the full-length TSH receptor

**Mihaly Mezei[1,2]\*, Rauf Latif[2,3], Terry F Davies[2,3]\***

[1]Department of Pharmacological Sciences, Icahn School of Medicine at Mount Sinai, New York, United States; [2]Department of Medicine, Thyroid Research Unit, Icahn School of Medicine at Mount Sinai, New York, United States; [3]James J. Peters VA Medical Center, New York, United States

**Abstract** (GPCR)The receptor for TSH receptor (TSHR), a G protein coupled receptor (GPCR), is of particular interest as the primary antigen in autoimmune hyperthyroidism (Graves' disease) caused by stimulating TSHR antibodies. To date, only one domain of the extracellular region of the TSHR has been crystallized. We have run a 1000 ns molecular dynamic simulation on a model of the entire TSHR generated by merging the extracellular region of the receptor, obtained using artificial intelligence, with our recent homology model of the transmembrane domain, embedded it in a lipid membrane and solvated it with water and counterions. The simulations showed that the structure of the transmembrane and leucine-rich domains were remarkably constant while the linker region (LR), known more commonly as the 'hinge region,' showed significant flexibility, forming several transient secondary structural elements. Furthermore, the relative orientation of the leucine-rich domain with the rest of the receptor was also seen to be variable. These data suggest that this LR is an intrinsically disordered protein. Furthermore, preliminary data simulating the full TSHR model complexed with its ligand (TSH) showed that (a) there is a strong affinity between the LR and TSH ligand and (b) the association of the LR and the TSH ligand reduces the structural fluctuations in the LR. This full-length model illustrates the importance of the LR in responding to ligand binding and lays the foundation for studies of pathologic TSHR autoantibodies complexed with the TSHR to give further insight into their interaction with the flexible LR.

**\*For correspondence:**
Mihaly.Mezei@mssm.edu (MM);
terry.davies@mssm.edu (TFD)

## Editor's evaluation

This valuable paper is methodologically solid as it describes the first molecular dynamics (MD) simulation of the full-length membrane-bound Thyroid Stimulating Hormone Receptor (TSHR). This paper will be of interest to researchers working on thyroid biology and autoimmune disorders. This important set of new results also highlights dynamic conformational changes in the linker region (LR) and its interaction with the leucine-rich domain (LRD). While most claims are convincingly supported by the data and advance the understanding of TSHR, the experimental validation is currently incomplete.

## Introduction

The TSH receptor (TSHR) on the surface of thyrocytes is an important regulator of thyroid growth, development, hormone synthesis, and secretion. It is also the primary target of autoantibodies in Graves' disease (autoimmune hyperthyroidism; *Davies et al., 2020*). From cloning, sequence analysis, partial crystallization, and biochemical studies, this GPCR has been deduced to be made of a large ectodomain (ECD) and membrane-bound signal transducing transmembrane domain (TMD; *Rapoport et al., 1998*; *Davies et al., 2005*). The ECD is further divided into a leucine-rich domain

(LRD) forming a curved structure which is linked to the TMD by a 130 amino acid (AA) linker region (LR) known commonly as the 'hinge region'" (AA280-410). Unique to the TSHR is a large 50 AA cleavage region (AA316-366) located within the LR that is proteolytically degraded leaving a cleaved ECD thought to be tethered to the TMD via three cysteine bonds (*Tanaka et al., 1999*; *Tanaka et al., 2001*) and sometimes referred to as the C peptide.

Stimulating TSHR antibodies only recognize the TSHR in its native form (*Latif et al., 2012*) indicating that the receptor-epitope must be in a conformation that the antibody can recognize. Crystallization studies (*Sanders et al., 2007*; *Sanders et al., 2011*), besides producing crystal structures for the LRD, have also shown that antibodies, which either stimulate or block TSHR signaling, only bind to the LRD when the receptor is conformationally correct and can compete for TSH binding. In contrast, 'neutral' antibodies to the TSHR which do not initiate a traditional signal (*Morshed et al., 2018*) nor inhibit TSH binding, predominantly, but not exclusively, bind to linear epitopes in the LR (*Sun, 2018*). Although the partial LRD structure has been determined with x-ray crystallography (*Miller-Gallacher et al., 2019*), no experimental structure has been found for the LR, and until recently, only partial models have been proposed (*Kleinau et al., 2013*; *Kleinau et al., 2017*; *Morshed et al., 2009*). On the basis of the immune response to the TSHR we, and others, have suggested that the LR is not an inert scaffold but rather an important ligand-specific structural and functional entity (*Schaarschmidt et al., 2014*), but its structure has not been examined in the context of the full-length receptor. However, the recent success of the artificial intelligence (AI)-based Alphafold2 (*Jumper et al., 2021*) program led us to believe that it might be possible to generate a full-length receptor structure by combining the LRD-LR

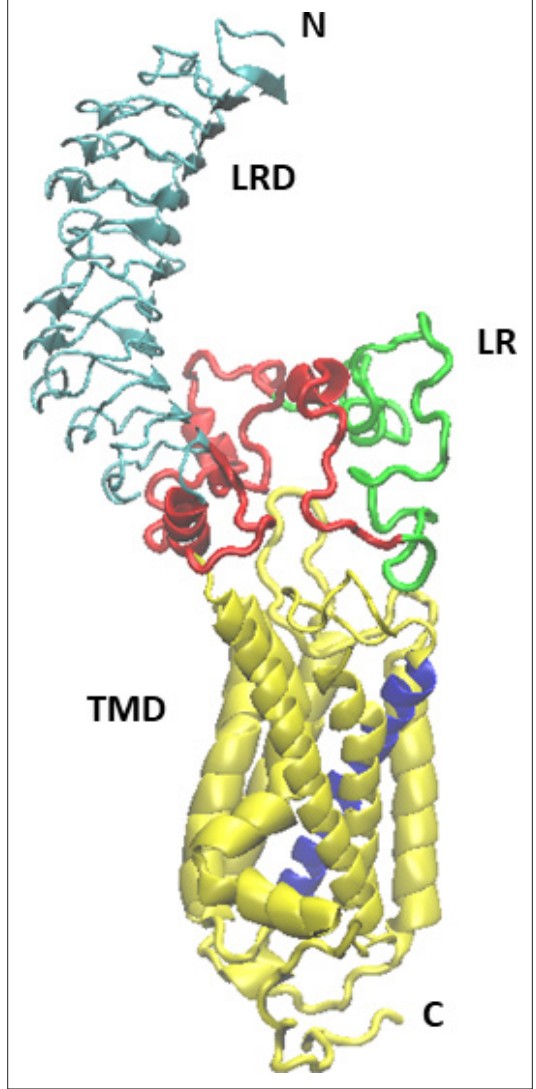

**Figure 1.** The initial model of the full-length TSH receptor (TSHR) (leucine-rich domain [LRD]: blue, linker region [LR:] red, and transmembrane domain [TMD]: yellow) derived from combination of the LRD and LR of the Alphafold2 program and the TMD of our earlier 'TRIO' model (*Mezei et al., 2021*). Helix 3 of the TMD is shown in purple.

structure generated by Alphafold2 (that includes a structural model of the LR region but no TMD and for which neither experimental nor homology models are available) with our recently published model of the TSHR TMD (named TRIO; *Mezei et al., 2021*). Note, that Alphafold2 provided a separate model for the TMD and did not provide a fully assembled structure. Therefore, we decided to use our own model as it was already equilibrated in the membrane environment and partially based on experimental validation. This full-length model could then be enhanced and verified with molecular dynamics (MDs) simulation. We can now report a successful computer-based approach to obtain insight into the LR allowing us to complete a full-length model of the TSHR. We have examined the behavior of this TSHR in a lipid-embedded, electro-neutral, aqueous environment by MD simulation studies and showed that the LR is indeed an intrinsically disordered protein but can be stabilized by TSH ligand binding to the LRD.

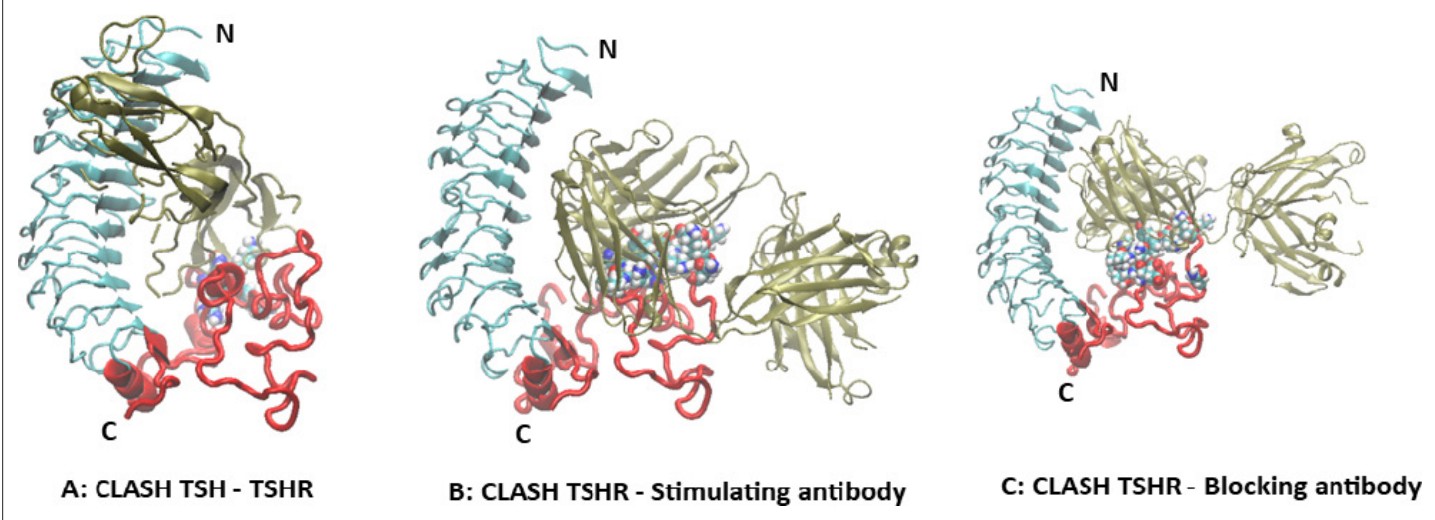

**A: CLASH TSH - TSHR**  **B: CLASH TSHR - Stimulating antibody**  **C: CLASH TSHR - Blocking antibody**

**Figure 2.** Initial model of the full-length TSHR receptor. (**A**) The extracellular part of the full-length model from *Figure 1* is shown in combination with the TSH ligand. The leucine-rich domain (LRD) region is shown in gray, the linker region (LR) backbone is shown in red, the ligand is green, and several LR residues clashing with the TSH are shown as spheres colored by atom types (partly obscured). For clarity, the transmembrane domain (TMD) has been removed in this and subsequent illustrations. (**B**) Similarly, the LR model is shown clashing with a stimulating TSH receptor (TSHR) monoclonal antibody (MS-1) based on the crystal structure (PDB id 3g04) with even more clashes than with TSH. (**C**) Here, the LR is clashing with a blocking TSHR monoclonal antibody (K1-70) based on the crystal structure (PDB id 2xwt) which once again shows many clashes.

## Results

### Initial full-length TSHR model and its clashes

*Figure 1* shows the structure of the full-length receptor that was obtained by combining the LRD-LR structure generated by the Alphafold2 AI system with the TMD structure from the TRIO model as detailed in Materials and methods section.

However, this assembled full-length TSHR structure showed significant LR clashes with TSH ligand and with TSHR antibodies (*Figure 2A–C*). A clash was here defined as a heavy atom distance less than 2.10 Â, 1.68 Â, and 1.65 Â, for atom pairs involving S, N or O, and C, respectively. In partic-ular, the number of LR heavy atoms clashing with TSH, blocking antibody (PDB id 2xwt), and stim-ulating antibody (PDB id 3g04) were 40, 68, and 101, respectively and involved 7, 11, and 18 resi-dues, respectively. Clearly, this approach showed marked hindrance of ligand and autoantibody binding indicating problems with the Alphafold2 model of the LR. We predicted that the MD simu-lation would be able to resolve these clashes and thus provide a refinement of the LR model.

The combined model, including Monte Carlo-generated internal waters, was then sent to the Charmm-gui server to be embedded in a dipal-mityilphospatidylcholine (DPPC) lipid bilayer and immersed in water with counterions. The membrane-inserted, fully hydrated, and neutral-ized system consisted of 177 and 176 DPPC mole-cules in the upper and lower layer, respectively, 148 $K^+$ and 149 $Cl^-$ ions and 54,412 water mole-cules, a total of 220,799 atoms in the simulation cell. The length of the periodic cell hexagon was 185.0 Â, and the edge of the base hexagon was 66.4 Â. *Figure 3* shows the full simulation cell.

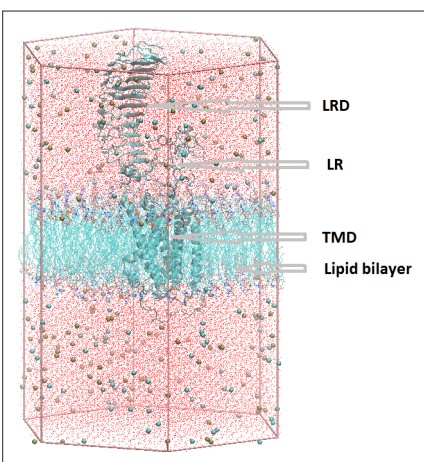

**Figure 3.** The full simulation cell prepared by Charmm-gui. The TSH receptor (TSHR) is shown in gray cartoon representation, lipids are shown as lines without hydrogens, ions as tan or cyan spheres representing $K^+$ or $Cl^-$ ions, respectively, and the water oxygens as red dots. The hexagonal prism edges defined the initial simulation cell.

As the structures of the LRD have been experimentally determined by crystallography and we have described the TSHR-TMD in detail earlier (*Mezei et al., 2021*), the analyses in this report are focused on extracting a potential structure of the entire LR from the MD simulation trajectory of the Alphafold2-based membrane-embedded structure.

Animation (with the VMD software) of the simulation trajectory showed that (a) the LRD and TMD structures in the simulated complex did not show significant deviation from the earlier reports; (b) the LR structure generated using the Alphafold2 program had few secondary structural elements (SSEs) and showed significant fluctuation; and (c) the relative orientation of the LRD with the rest of the protein also fluctuated significantly during the simulation. This, therefore, offered the opportunity for finding conformations where the possibility of ligand and antibody binding did not clash with the LR. *Figure 4A* shows the 2D root-mean-square deviation (RMSD) map of the LR over 2000 evenly spaced conformations. The RMSDs are calculated for the LR backbones. K-medoid clustering was performed asking for three clusters (as suggested by the 2D RMSD map), and the cluster representatives (the structure with the lowest maximum RMSD with the rest of the cluster members) were also extracted. These three representative structures of the LR are shown in *Figure 5* with the LR backbones of the three clusters in red and illustrating the unique 50 AA cleaved region in green along with their simulation times.

## Instability of the LR

Examination of these backbones clearly showed that the LR does not form a well-defined stable tertiary structure. The radius of gyration $R_g$, a measure of compactness, of the LR is shown in *Figure 4B* as a function of simulation time. It shows remarkable fluctuations with the range (the difference between the highest and lowest value) of $R_g$ values being 6.4 Å. In contrast, the range of $R_g$ values was only 1.6 Å for the larger LRD (not shown). The secondary structure of the LR was also tracked by the DSSP algorithm.

*Figure 6A* shows the SSEs found as the simulation progressed. Most SSEs are helices but, remarkably, in the 700–900 ns range several beta sheets formed and then dissolved while a helix at the N-terminal (residues 280–290) persisted throughout the calculations, *Figure 6A* also shows that all the other transient helices were seen to unwind or form only in the later stages of the simulation.

The history of hydrogen-bonded residue pairs for the LR is shown in *Figure 7*. Each line on the plot represents one residue pair. By this analysis, it was seen that the inter-domain hydrogen bonds between the LR and the LRD (THR250-VAL374, ALA252-VAL374, and LEU254-THR376) persisted throughout the simulation, although several of these residue pairs broke and reformed their hydrogen bonds during the run. This reflected the structural fluctuations similar to the fluctuations seen in the DSSP plot of the SSEs. Note, however, that these hydrogen bonds are not the ones creating most of the SSEs.

## Receptor orientation

During these studies, the relative orientation of the LRD with the TMD was also found to undergo significant fluctuations. *Figure 8A* demonstrates this flexibility by showing the conformation of the LRD and LR of the full-length model at 250 ns intervals, superimposed on the initial TMD backbone (without the C-terminal tail). *Figure 8B* shows the fluctuation of the angle between the Z axis and the first principal axis of the LRD. It is also clear from the figures that the rotation of the LRD with respect to the LR is largely confined to one axis. Note also that all LRD conformations stayed well within the simulation cell. *Figure 8A* also shows the wide range of conformations that the LR forms during the simulation and the changing shape of the 50 AA cleavage region (shown in green) within the LR which is reported to be cleaved by membrane bound matrix metalloprotease (*de Bernard et al., 1999*; *Tanaka et al., 2000*) both in the native and activated states of the receptor (*Latif et al., 2004*). Near the end of the simulation, the two end residues of this 50-residue segment (which will form the C peptide) become close ($C_\alpha$ - $C_\alpha$ distance is 6.3 Å) – which may be of significance for post-cleavage processing.

## Analysis of the TMD helix bundle

As part of the full-length receptor structure, we also carried out analyses of the constitutive variation in the helix geometry of the 2D RMSD map of 2000 structures. This was calculated based on

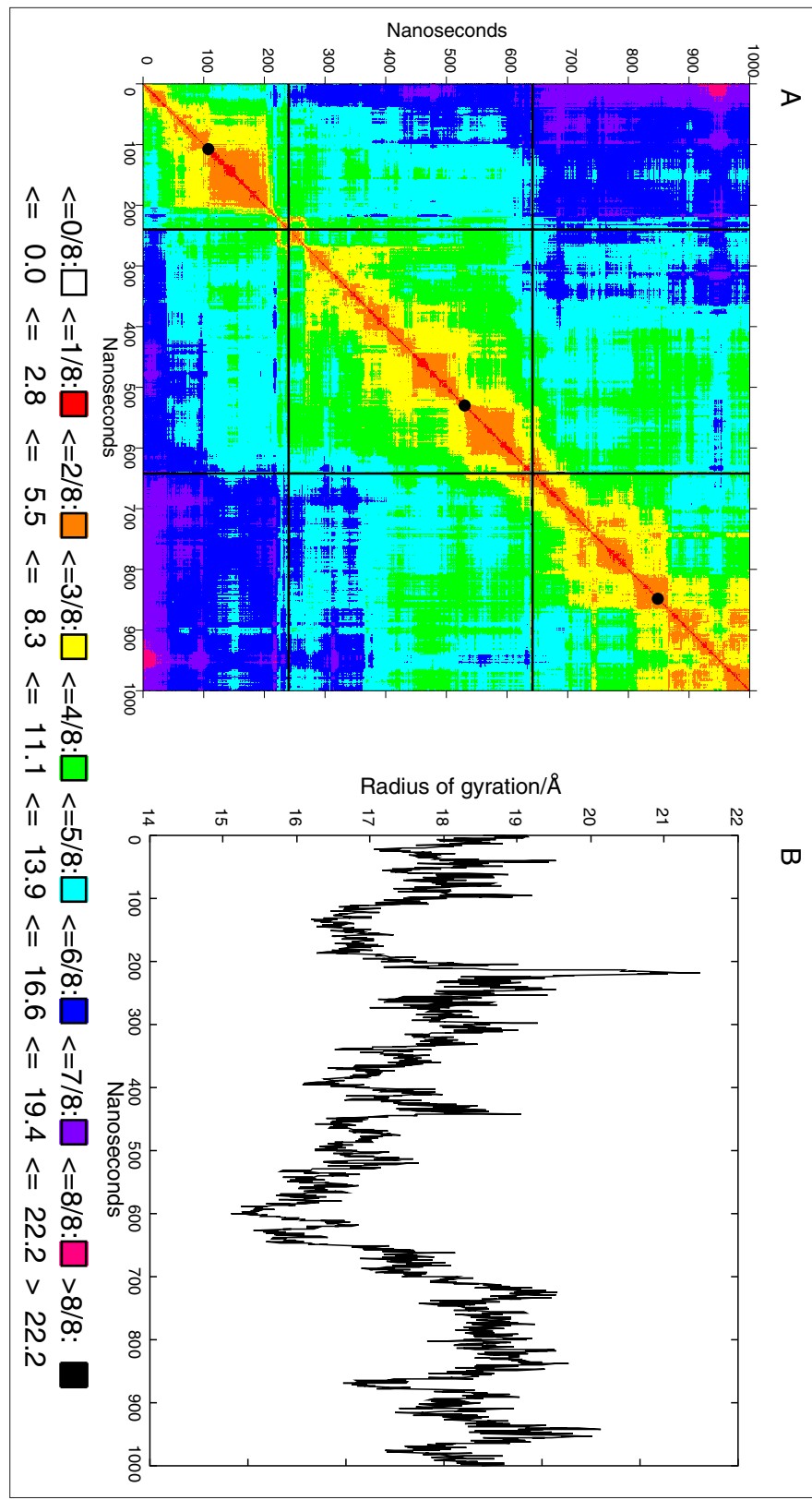

**Figure 4.** 2D RMSD map of the linker region (LR) during the 1000 ns simulation. (**A**) RMSD is in Å. Black lines delineate the three clusters, and the black discs on the diagonal indicate the most representative structure. (**B**) The radius of gyration (in Å) of the LR during the simulation.

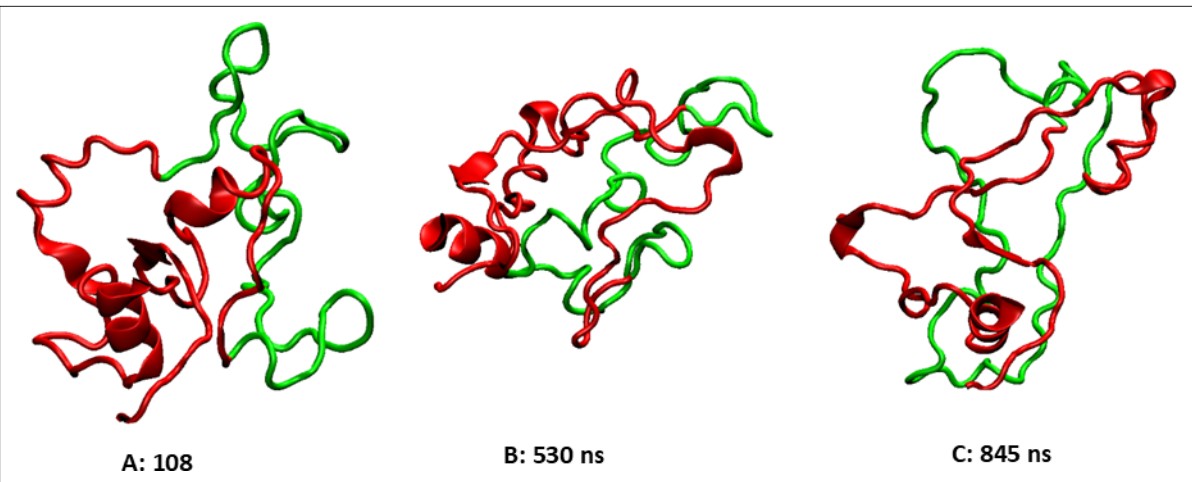

**Figure 5.** These three clusters are representative of the highly flexible structures of the linker region (LR) backbone at different times during the 1000 ns simulation. The 50 amino acid (AA) cleaved segment is shown in green, the rest of the LR is in red.

superimposing the structures on the TMD only as performed earlier. Using k-medoid clustering into three clusters resulted in three representative structures. The changes in the transmembrane helix bundle with respect to the initial model (whose structure was obtained from the third model of the previously published TRIO model; *Mezei et al., 2021*) were analyzed with the TRAJELIX (*Mezei and Filizola, 2006*) module of Simulaid. This program is based on the geometry of the $C_\alpha$ atoms, defining the helix axis (*Mezei, 2003*). Helices with proline are broken up into sub helices; the short segment between the two sub helices in helix 7 is ignored. Data in *Table 1* showed the change in the helix length and in the radius of the circle fitted to the $C_\alpha$ atoms, a measure of the bend of the helix (the smaller it is, the more bent is the helix). The change is the average over the representative structures minus the reference structure's value. When the reference value is outside the range of the values from the three representative structures, the change is deemed significant. The largest change was observed in helix 3 that became more curved, resulting in significant shortening (defined as the end-to-end distance).

Changes in the distance between the helix centers and the change in the closest approach of the helix axes are shown in *Table 2*. The comparison of the two values gives an indication of the relative shifts. The changes in the helix-helix angles are shown in *Table 3*. We noted from these data that, while the overall arrangement of the helix bundle did not change, it was clear that non-trivial adjustment of the helix bundle occurred. The extent of changes was similar to the changes we observed when the homology model was compared with representative structures from our earlier MD simulation of the TSHR TMD (*Mezei et al., 2021*).

### Analysis of the cysteines and cysteine bonds

The LR has six cysteines that are able to form three cysteine bonds: C283-C398, C284-C408, and C301-C390. In fact, in the Alphafold2 structure, the distance between the corresponding SG atoms are 3.44, 5.08, and 7.72 Å, respectively. Since the simulation did not include these bonds, it was interesting to see if the LR preferred conformations favorable for the cysteine bonds to form. *Figure 9A* shows the distances for the three putative bonds with time, and *Figure 9B* shows the position of the sulfur atoms in these cysteines (colored to match the corresponding graph color) in the Alphafold2 model of the LR. It is clear that C283 and C398 stayed consistently close and C284 and C408 did not separate too far from a bonding distance. However, the third pair, which actually were not close even in the Alphafold2 structure, quickly separated and never became close.

### Simulation of the TSHR in complex with TSH

Since our conclusion was that the LR is an intrinsically disordered protein (IDP), it was of interest to find out what, if anything, stabilizes its structure. The most likely candidate was its ligand, TSH. Thus, the structure shown in *Figure 10A* was used to set up an MD simulation modeled after the earlier

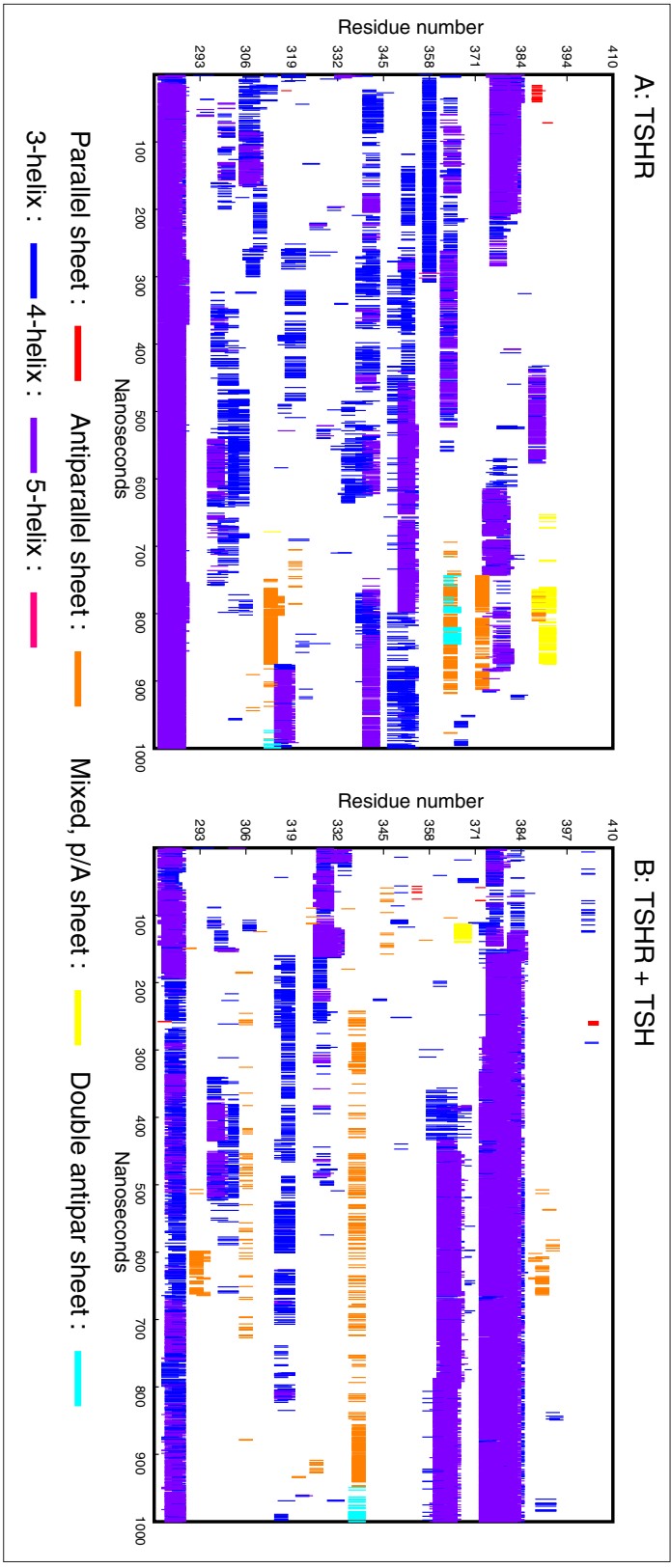

**Figure 6.** DSSP plots. (**A**) DSSP plot showing the secondary structure elements formed in the linker region (LR) during the simulation of the TSH receptor (TSHR) without ligand. The X axis is the simulation time, and the Y axis is the residue number. (**B**) DSSP plot showing the secondary structure elements formed in the LR during the simulation of the TSHR-TSH complex.

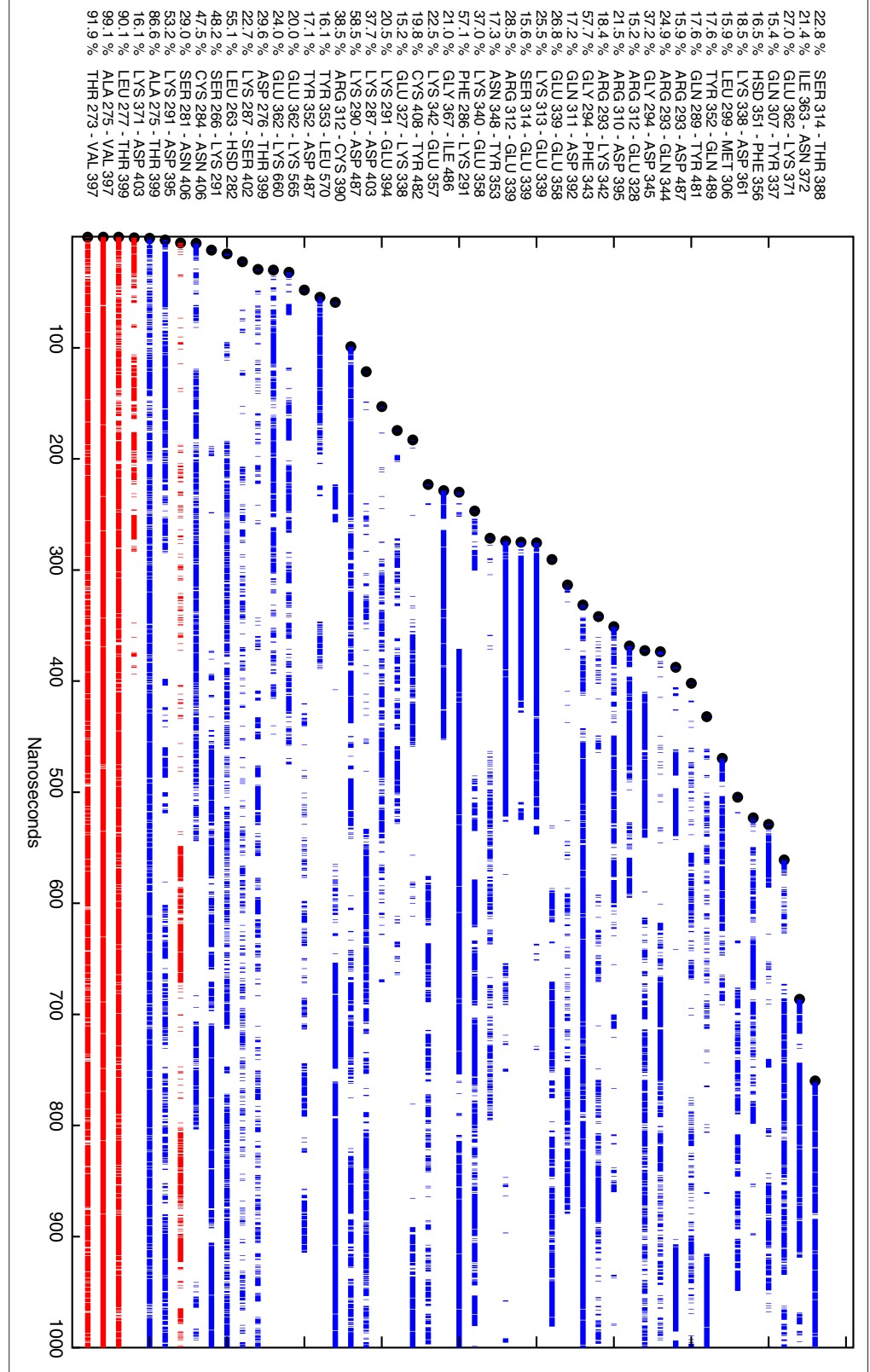

**Figure 7.** Plot of the residue pairs involving just the linker region (LR) that were hydrogen bonded at some parts of the simulation. The lines are broken whenever the residue pair was not hydrogen bonded. Blue represents residue pairs within the LR, and red represents hydrogen bonds between residues in the LR and the leucine-rich domain (LRD). Note the unbroken lines between the LR and LRD while the LR itself is intrinsically unstable. Note: residue

*Figure 7 continued on next page*

*Figure 7 continued*

pairs have to be at least five residues apart (to exclude the many intra-helix hydrogen bonds) and be hydrogen-bonded at least 15% of simulation time to be represented.

simulation without the TSH. The DSSP plot of the simulation is shown in *Figure 6B*. The SSEs were remarkably more stable, indicating that the presence of TSH indeed stabilized the LR. The conformations of the complex at the start, middle, and end of the 1000 ns simulation are shown in *Figure 10*. It clearly shows that the LR is attached to the TSH at several places. The hydrogen-bond analysis analogous to the one shown in *Figure 7* showed that there are seven residues which are hydrogen bonded to the α subunit of TSH and three LR residues hydrogen bonded to the β subunit, and there is even one LR residue that is hydrogen bonded to the LRD. It also shows that 6 of the 10 LR-TSH contacts were formed with the LR residues which are not part of the 50 AA cleaved segment. In addition, new contacts formed between the LR and the LRD with time.

It should be noted that the large number of contacts between TSH and TSHR may look surprising since the FSH-FSH receptor (FSHR) crystal structure (PDB id 4ay9) showed few, if any, contacts. However, that structure lacks precisely the region of the LR that is seen to be in contact with the ligand in the case of the TSHR. Also, the sequence alignment of the LR of TSHR and FSHR shows only 39.3% sequence identity. Furthermore, the large conformational fluctuations displayed by the LR exclude the possibility that the LR-TSH contacts are due to the bias of the initial conformation.

On the other hand, the LR moved significantly away from the TMD (more at 500 ns than at 1000 ns) indicating that to be able to transmit the signal induced by binding of TSH the cysteine bonds have to

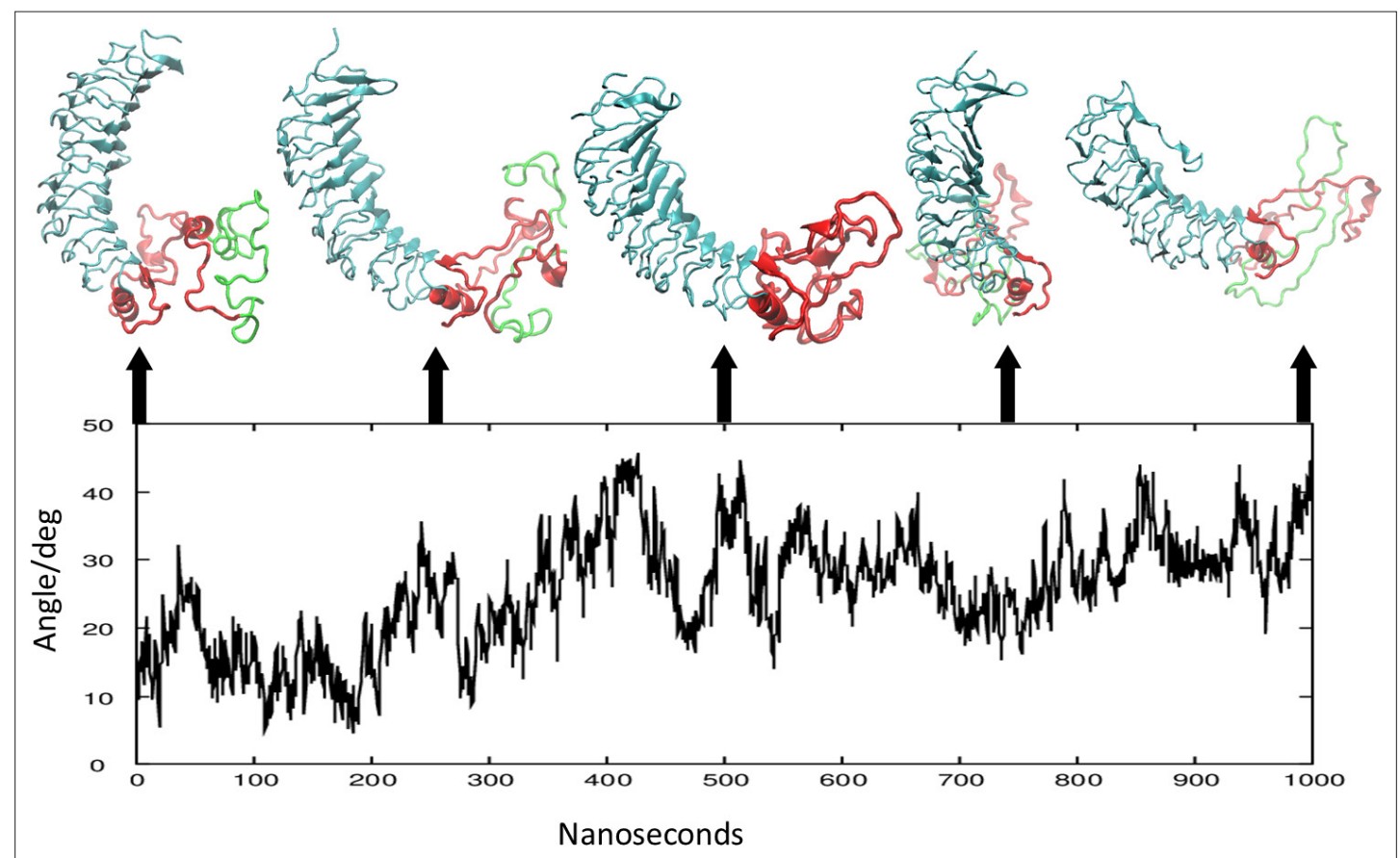

**Figure 8.** Cystein-cystein distances in the linker region (LR). (**A**) Comparison of the relative orientation of the leucine-rich domain (LRD) with the transmembrane domain (TMD) at 250 ns intervals. The structures are aligned by the TMD that is not shown. The linker region (LR) is shown in red with the 50 AA unique insert (316-366) that may be cleaved during post-translational processing is shown in green. (**B**) The instability of the LR is further illustrated by changes in the angle (in degrees) between the first principal axis of the LRD and the Z axis, over 1000 ns.

**Table 1.** Changes in helix length and radius.

| Helix # | 1 | | 2 | | 3 | | 4 | | 5 | | 6.1 | | 6.2 | | 7.1 | | 7.2 | | 8 | |
|---|---|---|---|---|---|---|---|---|---|---|---|---|---|---|---|---|---|---|---|---|
| Length: | −0.6 | S | −1.3 | S | −4.4 | S | 1.2 | S | 0.1 | n | 0.0 | n | 0.5 | S | 0.5 | n | −1.4 | S | −1.6 | S |
| Radius: | −0.5 | S | −0.5 | S | −3.7 | S | 0.4 | n | 0.2 | n | 0.0 | n | 0.2 | S | 0.5 | S | −0.4 | S | −0.4 | S |

Changes were defined as the difference between the average of values from the representative structures and from the starting model structure.

A positive number indicates an increase with respect to the starting structure. The characters 'S' and 'N' indicate that the reference value is within or outside the range of the representative structure values, respectively. The labels of the proline-separated segments of helices 6 and 7 have 0.1 and 0.2 added.

be present in the LR. While such analyses remain preliminary and subject to more and possibly longer simulations, the structure-inducing effect of the TSH appears to be very clear.

## Discussion

The TSHR, similar to the FSH and LH/hCG receptors, consists of LRD, a large extracellular ligand binding domain incorporating 11 repeats, and a TMD, linked via a 130 AA LR. The TMD is made up of eight helices joined by extracellular and intracellular loops and a long C-terminal cytoplasmic tail. The TMD is embedded in a phospholipid bilayer and transduces a cascade of signals by engaging several different G proteins (*Laugwitz et al., 1996*) and β arrestins (*Boutin et al., 2014*; *Frenzel et al., 2006*). The interest in the TSHR has been largely fueled by its role as a major human autoantigen in autoimmune thyroid disease, especially Graves' disease (*Davies et al., 2020*).

Detailed mapping of binding sites and interaction partners for the TSHR ligand, TSH, and for stimulating and blocking monoclonal autoantibodies to the TSHR, have been revealed by homology modeling (*Núñez Miguel et al., 2004*) and crystallization of the partial ECD bound to these autoantibodies (*Sanders et al., 2007*; *Sanders et al., 2011*). Furthermore, homology modeling has suggested possible mechanisms by which activation of the receptor by TSH and a stimulating antibody might occur (*Jiang et al., 2014*; *Kleinau and Krause, 2009*; *Krause et al., 2012*). However, all these tripartite models have remained incomplete due to the lack of a reasonable structure for the large TSHR

**Table 2.** Helix-helix distance changes.

| Helix# | 1 | | 2 | | 3 | | 4 | | 5 | | 6.1 | | 6.2 | | 7.1 | | 7.2 | | 8 | |
|---|---|---|---|---|---|---|---|---|---|---|---|---|---|---|---|---|---|---|---|---|
| 1 | 0.0 | n | −0.4 | S | 0.1 | n | −0.1 | n | 2.2 | S | 1.5 | S | 1.9 | n | −5.0 | S | 0.2 | n | −0.4 | n |
| 2 | 0.1 | n | 0.0 | n | 0.4 | S | −1.6 | S | 1.3 | S | −0.9 | S | −0.4 | n | 3.8 | S | 0.8 | n | −5.4 | S |
| 3 | 1.1 | S | 0.7 | S | 0.0 | n | 0.1 | n | −0.4 | n | −1.0 | S | −1.1 | n | 2.3 | n | 7.0 | n | −10.1 | S |
| 4 | 0.5 | n | 0.3 | S | 0.2 | S | 0.0 | n | 0.7 | S | 3.8 | S | −1.3 | S | 8.2 | S | 8.5 | S | −7.2 | S |
| 5 | 2.8 | S | 1.2 | S | −0.2 | n | −0.2 | n | 0.0 | n | −2.0 | S | 0.0 | n | 2.5 | S | 3.2 | n | −11.2 | S |
| 6.1 | 2.7 | S | 0.5 | n | −1.2 | S | −1.1 | S | 0.1 | n | 0.0 | n | −0.8 | S | −1.1 | S | −3.6 | n | −6.9 | S |
| 6.2 | 1.0 | S | −0.8 | S | −1.5 | S | −1.0 | S | 0.1 | n | 0.0 | n | 0.0 | n | −0.3 | n | −2.1 | S | 1.3 | n |
| 7.1 | 2.0 | S | 1.2 | S | 1.2 | S | 1.4 | S | 1.6 | S | 0.7 | S | −0.5 | S | 0.0 | n | 1.0 | n | −7.3 | S |
| 7.2 | 1.9 | S | −2.3 | S | −2.6 | S | −3.7 | S | −0.1 | n | 0.5 | n | 0.2 | n | 1.6 | S | 0.0 | n | −0.9 | S |
| 8 | 1.8 | n | −0.5 | n | −0.8 | S | −2.6 | S | 1.2 | S | 1.9 | S | 0.8 | S | 2.4 | S | 0.3 | n | 0.0 | n |

Upper triangle shows the change in the closest approach of the helix axes; the lower triangle shows the change in the distance between the helix centers. Positive number indicates an increase with respect to the starting structure. The characters 'S' and 'n' indicate that the reference value is within or outside the range of the representative structure values, respectively. The labels of the proline-separated segments of helices 6 and 7 have 0.1 and 0.2 added.

**Table 3.** Helix-helix angle changes.

| Helix # | 2 | | 3 | | 4 | | 5 | | 6.1 | | 6.2 | | 7.1 | | 7.2 | | 8 | |
|---|---|---|---|---|---|---|---|---|---|---|---|---|---|---|---|---|---|---|
| 1 | 0.9 | n | 1.4 | S | −7.8 | S | 6.9 | S | −10.8 | S | −8.8 | S | 16.4 | S | −27.8 | S | 11.8 | n |
| 2 | | | −0.3 | n | −1.7 | S | 0.2 | n | −0.6 | n | −0.8 | n | −8.6 | S | 34.8 | S | −5.8 | n |
| 3 | | | | | 2.4 | S | −2.3 | S | 7.5 | S | 1.6 | n | 6.6 | n | −24.2 | S | 1.0 | n |
| 4 | | | | | | | −0.2 | n | 5.3 | S | 1.1 | n | −3.3 | n | 34.3 | S | −4.1 | n |
| 5 | | | | | | | | | 4.1 | S | −0.9 | n | 4.5 | n | −24.5 | S | 0.1 | n |
| 6.1 | | | | | | | | | | | 5.5 | S | −9.0 | n | 36.1 | S | −7.4 | S |
| 6.2 | | | | | | | | | | | | | 4.1 | n | 0 8.9 | S | 2.9 | n |
| 7.1 | | | | | | | | | | | | | | | −31.9 | S | 0.5 | n |
| 7.2 | | | | | | | | | | | | | | | | | −10.7 | S |

The characters 'S' and 'n' indicate that the reference value is within or outside the range of the representative structure values, respectively. The labels of the proline-separated segments of helices 6 and 7 have 0.1 and 0.2 added.

LR/hinge region. Here, we present a full-length model of the TSHR which became possible because of the recent availability of models of the human proteome generated by the AI-based protein folding program Alphafold2 (*Jumper et al., 2021*). We combined the Aplhafold2-generated structure of the LRD-LR complex with our recent MD-refined homology model of the TSHR TMD (*Mezei et al., 2021*) and further refined the structure with MDs in a DPCC membrane environment. Given that the structures of the LRD and the TMD of the TSHR have been described in earlier studies (*Ali et al., 2015*; *Latif et al., 2015*), our analysis in this report was firstly focused on the structure of the LR, its intramolecular and molecular bonding dynamics, and its structural variations in lieu with LRD and TMD structures.

It is notable that in the initial structure obtained with our heuristic process by the combination of the two parts using the relative orientation of the LRD (residues 22–280) and the TMD (residues 409–694) resulted in a full-length receptor model that retained the relative orientation of the ECD and the TMD in spite of the large conformational fluctuation of the LR (*Figure 5*) and the large fluctuation on the angle between the LR and the LRD (*Figure 8*) supporting the overall correctness of our initial model. Further support can be seen in the fact that the relative orientation of the ECD and the TMD allows the formation of TMD dimers in a conformation predicted and experimentally verified by our earlier work (*Latif et al., 2015*) and which would still leave the LRD-binding surface free for ligand or auto-antibody binding. However, in the Alphafold2-predicted conformation presented here, the concave surface of the LRD where autoantibodies bind, is partly occluded by the LR and thus would clash with a bound TSHR antibody whose binding sites on the LRD are known (PDB id 3g04). This conundrum was resolved by the observations that (a) the structure of the LR cluster is highly flexible and (b) the relative orientation of the LRD with the rest of the structure is also highly variable resulting in a significant population of conformations where stimulating and blocking TSHR autoantibodies could access the receptor to activate or block signaling. Thus, we can say that the Alphafold2 structure of the LR, while useful in providing a starting point for the MD simulation, was not correct in the sense that it missed the large conformational freedom of the LR needed for ligand binding. Given the low reliability score assigned to the LR part of the Alphafold2 structure, this observation was not a surprise. We also submitted the LR sequence to the PONDR (Predictors of Natural Disordered Regions) server (http://www.pondr.com/) – using the methods VLXT (the default), VL3 NNP, and VL2 NNP, and it predicted that 28.91%, 64.81, and 62.5%, respectively, of the LR is disordered. Furthermore, these conclusions could be further verified by obtaining, if possible, cryo electron microscopy (CEM) or a crystal structure of the native full-length TSHR perhaps stabilized by an autoantibody to the LR.

Based on this study, we can conclude that the LR in the TSHR is remarkably flexible, sampling vastly different overall conformations without settling on a well-defined tertiary structure. While different SSEs formed during the simulation (mostly helices), they were transient, as seen from *Figure 6A*. On the other hand, contacts between the LR and the LRD persisted throughout the simulation. The

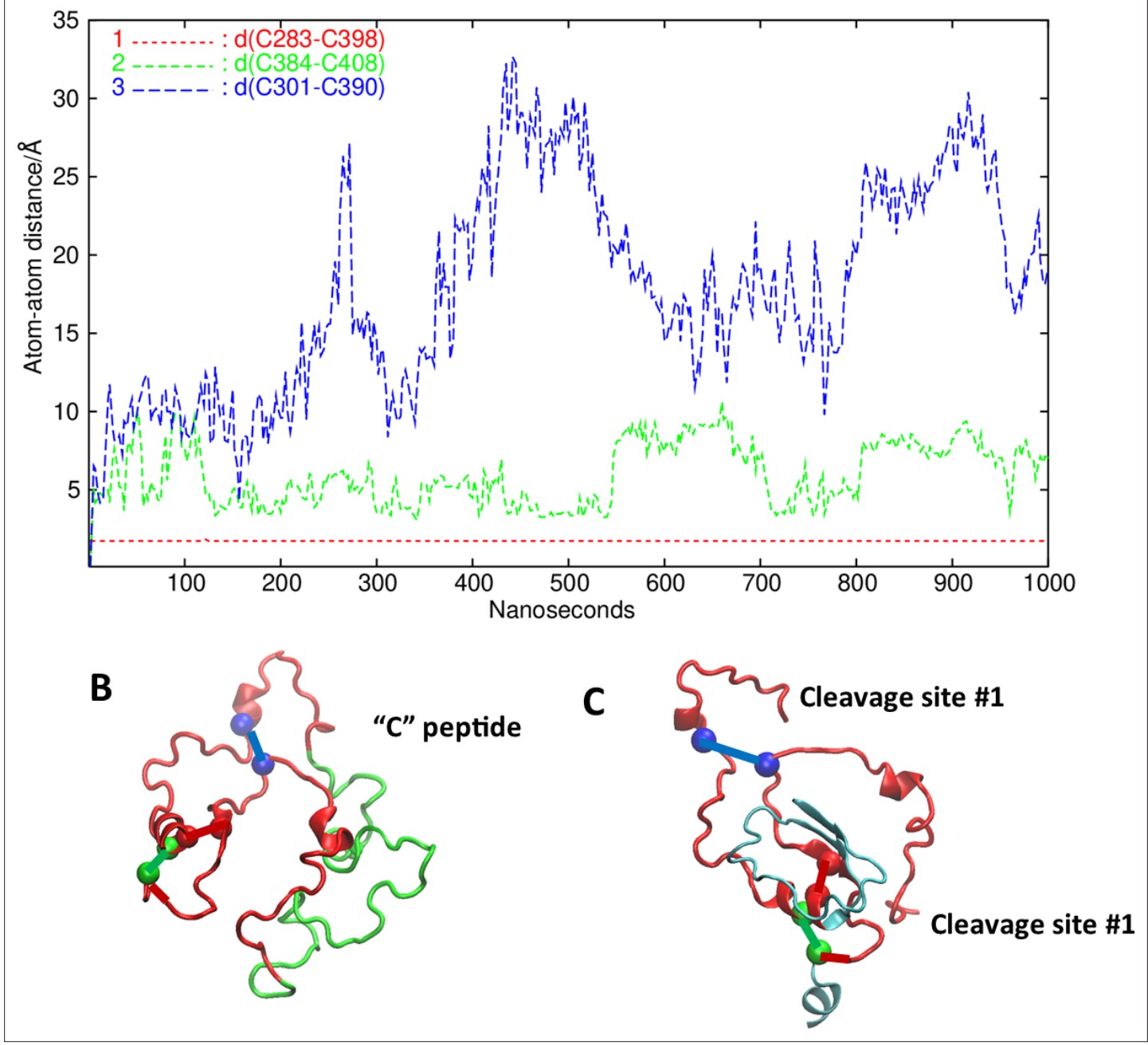

**Figure 9.** Analysis of putative disulfide bonds. (**A**) Time evolution of the three cysteine-cysteine distances (in Å) in the linker region (LR). C283-C398: red, C384-C408: green, and C301-C390: blue. (**B**) The LR backbone (red and green) and the S atoms of the cysteines, colored to match the corresponding graph color. The putative pairs are connected by a line. (**C**) Here, we have left the cysteine pairs connected but taken away the 50 amino acid (AA) insert in the LR and show the reported cleavage sites. For giving context, a small part of the leucine-rich domain (LRD) and the transmembrane domain (TMD) are also shown in gray/blue.

conclusion is that the LR is an IDP. This conclusion is consistent both with the fact that so far no one has succeeded in obtaining its crystal structure and with the suggestion (*Jumper et al., 2021*) that low reliability scores indicate that the protein is intrinsically disordered. However, from these studies, we speculate that the highly flexible nature of the LR is what allows it to accommodate both the ligand and the autoantibodies to the LRD.

The fact that the two pairs of cysteines stayed close with their putative partners for disulfide bonding indicated that these cysteines should form a disulfide bond. Furthermore, the fact that when

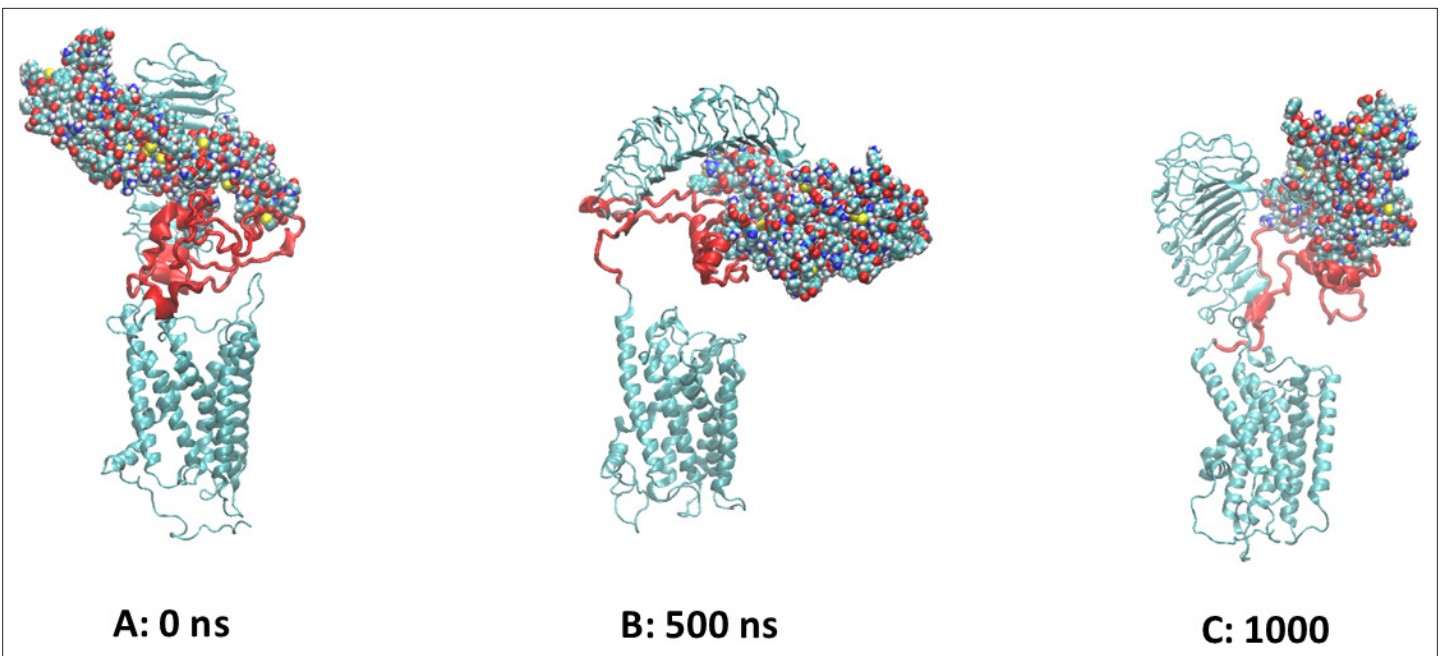

**Figure 10.** The conformations of the TSH receptor (TSHR)-TSH complex at the start, middle, and end of the simulation. TSH is shown as spheres, the linker region (LR) backbone is red, and the leucine-rich domain (LRD) and transmembrane domain (TMD) are gray/blue.

the TSH was added to this model, these cysteines moved apart also suggested that they could be part of the signal-transduction mechanism.

The convergence of the simulation is always an open question. However, there are important indicators that suggest that our sampling was adequate. *Figure 7*, the history of hydrogen bonds involving the LR, showed that most of the bonds had formed during the first half of the simulation; no new ones formed in the last quarter. Hydrogen bonds also kept forming and reforming. Similarly, *Figure 6A* shows that most SSEs formed and broke several times during the simulation. Taken together, these indicators allow us to conclude that the simulation involved adequate sampling.

In addition to our examination of the LR structure, we compared the changes seen in the TMD helix bundle, from the respective reference structure in the TMD-only simulation, versus the full-length model. It was of great interest that the change in the curvature (and, as a consequence, in the end-to-end distance) of helix 3 was significantly greater in the new full-length model than in the TMD-only model. Comparing the range of values sampled in the representative structures (data not shown), we found similar differences. This observation supported the hypothesis put forward earlier (*Davies and Latif, 2015*) that helix 3 is highly important for the signal transduction of the TSHR and consistent with a variety of small molecule activators which all interact with helix 3 (not illustrated). We also examined the LR cysteines. Much has been discussed concerning the role of cysteine bonds in anchoring the LR to the LRD following post-receptor processing which involves cleavage of the unique 55 AA insert in the LR (see *Figure 9C*). The analysis of the cysteines in the LR showed the remarkable affinity of one pair to each other (C283-C398) and to a lesser degree for the second pair (C384-C408). However, the third pair (C301-C390) showed poor affinity, leading to the suggestion that in the fully formed TSHR, only two of the cysteine bonds were formed.

Recently, structures of the luteinizing hormone receptor (LHR) in complex with different proteins and in its inactive state, a total of five structures, obtained by CEM have been described (*Duan et al., 2021*). In accordance with our simulation, the relative orientation of the LRD with the TMD was quite variable. The LR in the LHR, however, while significantly shorter than the LR in the TSHR, was missing more than 40 residues in the CEM structures, thus preventing detailed comparison with the TSHR LR. It is notable that the short helix at the N-terminal end of the LR was present in all five CEM structures and during the 1000 ns MD run described here. Also, the LR conformations in the CEM structures were very different from each other, reflecting the conformational variations observed during our TSHR MD run. Interestingly, our simulation of 1000 ns with the heterodimeric TSH showed stabilization of SSEs,

which was clearly absent without TSH (*Figure 6*), further suggesting that the LR plays an important part in TSH action and that TSH is able to stabilize the flexible LR region after transitioning through various dynamic structural changes. Furthermore, the separation of the LR and the TMD during the simulation suggests that for the proper action of the TSHR on binding of the TSH, then at least one cysteine bond has to be formed.

Our simulation of the full-length TSHR embedded in a lipid membrane, solvated with water-containing counterions, i.e., in a biologically relevant environment suggested (a) the LRD and the TMD continue to maintain their fold; (b) the LR is flexible, but maintains protein-like behavior forming secondary structure elements that are, however, transient; and (c) the relative orientation of the LRD is also variable. Both the orientation of the LRD and the structural flexibility of the LR suggests that these features are likely to be important for the TSHR to accommodate the diverse ligands such as TSH and autoantibodies that are known to bind to the extracellular region of TSHR.

Our preliminary results from a simulation of the TSHR complexed with its ligand TSH showed that TSH formed strong interactions with both the LRD and the LR. Furthermore, it appeared to increase the longevity of the secondary structure elements in the LR. In addition, the breakup of the LR-TMD interface points to the importance of the disulfide bonds in the LR.

In conclusion, we performed the first MDs simulation of the full-length TSHR that includes the characterization of a flexible LR and concluded that the LR is constitutively unstable in the native state of and receptor thus can be considered an IDP. The earlier generated models (e.g. *Núñez Miguel et al., 2004*) only represented one conformation of the many LR structures as well as missing the 50 AA insert unique to the TSHR and also highlights the importance of considering conformational ensembles as the most accurate model of the LR. In addition, our preliminary results indicate that TSH is able to stabilize this disordered protein, which suggests that stabilization of the LR is important for signaling to ensue.

## Materials and methods
### AI model of the LRD and LR
The coordinates of the structure of the LRD and LR domains of the human TSHR, residues 24–408, were downloaded from the Swissprot database (*Bairoch and Apweiler, 2000*); Uniprot #: P16473 and Swissprot file /P1/64/73. The first 23 residues, of which 21 residues formed the signal peptide, were not included in the model. The downloaded structure was used without any modification (apart from translation and rotation) in the initial model, available from the Dryad server (vide infra).

### Model of the TMD
Our previous work (*Mezei et al., 2021*) detailed the MD trajectory of the TMD, residues 408–717 of the human TSHR, into three clusters using k-medoid clustering (*aPJR, 1987*), performed by the program Simulaid (*Mezei, 2010*). For the present work, we chose the representative structure of the largest cluster, forming during the second half of the MD trajectory. As before, the initial positions of internal waters were determined using grand-canonical ensemble Monte Carlo simulation (*Mezei, 1987*), followed by circular variance (*Mardia and Jupp, 1999*) filtering and derivation of generic sites (*Mezei and Beveridge, 1984*). The Monte Carlo simulation, as well as the circular variance and generic site calculations, were performed with the program MMC (*Ali et al., 2015*).

### Formation of a full-length TSH model by combination of the Alphafold2 LRD and LR model with the TRIO TMD model
The Alphafold2 model of the TSHR ECD (LRD-LR) and the TRIO model have only one common residue – cysteine 408. First, the LRD-LR model was translated so that the $C_\alpha$ of the LRD-LR cysteine is at the position of the TRIO cysteine's $C_\alpha$ position. As both models were already oriented along the membrane normal (the Z axis), the only degree of freedom left was rotation around the Z axis. In the next step, a scan by 45° steps selected the angle region that minimized the volume of the enclosing rectangle, followed by generating conformations in 5° steps and obtaining the list of contact distances between the LR and the TMD. Pairs of atoms, e.g., $i_A$ of domain $A$ and $j_B$ of domain $B$, are defined to be in contact if they are mutually proximal. This means that atom $i_A$ is the nearest atom in domain $A$ to atom $j_B$, and atom $j_B$ is the nearest atom in domain $B$ to atom $i_A$ (*Mezei and Zhou, 2010*). Examination

of the contacts narrowed down the likely conformation. The final choice was made after having examined visually (using the program VMD; *Humphrey et al., 1996*) the form that resulted in the broadest contacts between the LR and the TMD. While this last step is admittedly an inexact operation, it is made with the understanding that small errors would be corrected during the MD equilibration. The coordinates of this initial model are available from the Dryad server at the URL https://doi.org/10.5061/dryad.rjdfn2zdp.

## Immersion in bilayer

The Charmm-gui server (*Jo et al., 2008*) was used to immerse the full model of TSHR, including the internal waters, into a bilayer of DPPC molecules. The server also added a water layer as well as counterions ($K^+$ and $Cl^-$ ions), both to ensure electroneutrality and an ionic strength of 0.15 M to best represent physiological conditions. Periodic boundary conditions were applied using a hexagonal prism simulation cell. The system thus generated included inputs for a six-step equilibration protocol (*Wu et al., 2014*) and inputs for the production run, all using the program NAMD (*Phillips et al., 2005*).

## MD simulation

The simulations used the default parameters set by Charmm-gui. For the protein and the ions, the pairwise additive Charmm36m force field (*Huang et al., 2017*) was used, and water was represented by the TIP3P water model (*Jorgensen et al., 1983*). Long-range electrostatics was treated with the Ewald method, and VdW interactions used a cutoff of 12 Â, smoothly cut to zero starting at 10 Â. The simulations used 2 fs time steps and were run in the (T, P, N) ensemble.

## TSHR-TSH and TSHR-antibody complexes

The TSHR-TSH complex used for calculating clashes with the Alphafold2 model and for the start of the MD simulation was obtained based on the crystal structure of the FSH in complex of the LRD of FSH (PDB id 4ay9). In the first step, the several TSH beta chain coordinates were generated based on the FSH beta chain coordinates in the FSHR-FSH complex using the program Modeller (*Webb and Sali, 2014*). Next, the model with that had the fewest clashes with the LRD was selected and used to replace the FSH beta coordinates, followed by aligning the LRD of the FSHR-TSH beta complex to the LRD of our full-TSHR model. Finally, the beta chain of one structure from an earlier unpublished model of the TSH-LRD complex was aligned to the beta chain of the newly generated complex to add the TSH alpha chain to the model.

The TSHR-antibody complexes for the calculation of clashes were obtained by superimposing the LRDs in the crystal structures of stimulating and blocking antibodies (PDB ids 3g84 and 2xwt) to the LRD of the Alphafold2 structure.

## Analyses

Most analyses were performed on the trajectories with the program Simulaid (*Mezei, 2010*). Hydrogen bonds are defined by Simulaid as X···H-Y where X and Y are polar heavy atoms, the X···H-Y angle is above 120° and the X-H distance is below threshold. The values used for N-H, O-H, P-H, and S-H thresholds were 2.52, 2.52, 3.24, and 3.15 Å, respectively. Note that this definition ignores the actual charges thus it includes salt bridges as well as is the case for several of the hydrogen bonds thus defined. The adequacy of the run length was verified by the saturation of the hydrogen-bond tracks. In other words, after a while the system did not form new hydrogen bonds, it only broke and reformed the existing ones. The variation of the shape of the LR was tracked by calculating the $R_g$. The change in the relative orientation of the LRD with the rest of the protein was characterized by the angle between the first principal axis of the LRD and the Z axis and also tracked by observing the animated trajectory. Formation and unraveling of SSEs in the LR were tracked with the DSSP algorithm (*Kabsch and Sander, 1983*).

## Acknowledgements

This work was supported in part by NIH grant DK069713 and a VA Merit Award BX000800 (to TFD), the Segal Family Fund and additional anonymous donations. It was also supported in part through the computational resources and staff expertise provided by the Department of Scientific Computing at the Icahn School of Medicine at Mount Sinai.

## Additional information

### Competing interests
Terry F Davies: TFD is member of the Board of Kronus Inc (Starr, ID, USA). The other authors declare that no competing interests exist.

### Funding

| Funder | Grant reference number | Author |
|---|---|---|
| NIH Office of the Director | DK069713 | Terry F Davies |
| Veterans Administration merit award | BX000800 | Terry F Davies |
| Segal Family Foundation | 00000000 | Terry F Davies |

The funders had no role in study design, data collection and interpretation, or the decision to submit the work for publication.

### Author contributions
Mihaly Mezei, Conceptualization, Software, Formal analysis, Methodology, Writing – original draft, Writing – review and editing; Rauf Latif, Conceptualization, Writing – review and editing; Terry F Davies, Conceptualization, Formal analysis, Funding acquisition, Writing – review and editing

### Author ORCIDs
Mihaly Mezei http://orcid.org/0000-0003-0294-4307
Rauf Latif http://orcid.org/0000-0002-4226-3728
Terry F Davies http://orcid.org/0000-0003-3909-2750

### Decision letter and Author response
Decision letter https://doi.org/10.7554/eLife.81415.sa1
Author response https://doi.org/10.7554/eLife.81415.sa2

## Additional files

### Supplementary files
• MDAR checklist

### Data availability
The initial model generated is available from the Dryad server; URL: https://doi.org/10.5061/dryad.rjdfn2zdp. The software used for the analysis are available at the URL https://mezeim01.u.hpc.mssm.edu.

The following dataset was generated:

| Author(s) | Year | Dataset title | Dataset URL | Database and Identifier |
|---|---|---|---|---|
| Mezei M, Latif R, Davies T | 2022 | TSHR full model | https://doi.org/10.5061/dryad.rjdfn2zdp | Dryad, 10.5061/dryad.rjdfn2zdp |

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
