## [Editor Report]

This valuable paper is methodologically solid as it describes the first molecular dynamics (MD) simulation of the full-length membrane-bound Thyroid Stimulating Hormone Receptor (TSHR). This paper will be of interest to researchers working on thyroid biology and autoimmune disorders. This important set of new results also highlights dynamic conformational changes in the linker region (LR) and its interaction with the leucine-rich domain (LRD). While most claims are convincingly supported by the data and advance the understanding of TSHR, the experimental validation is currently incomplete.

---

## [Decision Letter]

**Decision letter after peer review:**

Thank you for submitting your article "Computational model of the full-length TSH receptor" for consideration by *eLife*. Your article has been reviewed by 2 peer reviewers, and the evaluation has been overseen by a Reviewing Editor and Carlos Isales as the Senior Editor. The reviewers have opted to remain anonymous.

Essential revisions:

Reviewers' comments are appended. Please address all comments raised by the reviewers.

*Reviewer #1 (Recommendations for the authors):*

Major weaknesses:

– Page 29 line 508: The authors claim that this is the first model of full-length TSHR, but citation 39 appears to be a full-length TSHR homology model (with noted differences in LR conformation). This manuscript does appear to be the first simulation of full-length TSHR, however. The manuscript would benefit from stating more clearly what is in this model that previous models are lacking and why that is significant.

– Page 7 line 88: This paragraph describes the choice of orientation for the starting model. This model was selected first by rotating the LRD-LR around the z-axis and selecting an orientation that minimizes the volume of the box necessary for running the simulation and maximizes the number of contacts between LR and TMD (which the authors acknowledge as inexact). While this choice of starting model is computationally efficient, it is not sufficiently rigorous. The initial model appears to contain the interdomain hydrogen bonds that the authors track in Figure 7, and it is unclear that these bonds would be found and maintained if the starting conformation of the model were different. It is also unclear whether the TSHR-TSH interactions reported in Figure 10 would hold with a different starting conformation. To address this concern, the paper would benefit from another set of calculations where this arbitrary orientation is changed to be outside of the 45{degree sign} window that is presented to validate that these results are not dependent on initial orientation. It may also be valuable to report the orientation that AF2 produces (see next comment).

– The authors used AF2 to determine the structure of the whole protein except for the Transmembrane domain (TMD), for which they had generated a model previously. How similar was the AF2 model of the whole protein to their model of two AF2 domains with the TMD? What does the TRIO model of the TMD give that an AF2 model of the TMD doesn't? AF2 has been shown to produce reasonable models of membrane bound proteins (Diego del Alamo, Davide Sala, Hassane S Mchaourab, Jens Meiler (2022) Sampling alternative conformational states of transporters and receptors with AlphaFold2 *eLife* 11:e75751); it would be an important addition if the authors could describe how/why AF2 cannot be used for this entire structure.

– It would be helpful if the authors could expand on the signaling mechanism mentioned in the last line of the discussion. This result and its mechanistic implications may be important in further studies (both computational and experimental) but it is barely discussed here.

– Page 8 line 106: MD parameters necessary to run the six-step equilibration protocol and production run should be reported.

– Page 10 line 156: This paragraph notes that many atoms clash and that the model requires further development. However, there is no mention of what was done to further develop the model. Any steps that were taken to further develop should be reported.

– Page 7 line 92: The authors should provide a rigorous definition of "mutually proximal". A distance threshold would be sufficient.

– Page 9 line 135: the X-Y distance is below what threshold for hydrogen bonds?

*Reviewer #2 (Recommendations for the authors):*

1. The authors should put more effort to provide clear, properly labeled and good quality figures.

a. There is no labeling of figures like N-, C- terminal of the protein in Figure 2.

b. In Figure 4a, its very difficult to read the color codes below the figure.

c. For fig6, I can hardly read the labels on y-axis.

d. There is typo in line 364. it should be 284 and not 384.

2. In Figure 1, the authors showed the enlarged LR showing various epitopes. What is the purpose of this?

3. The authors have used the pdb code 4AY9 for the structural analysis. They should also use sequence based analysis and find out whether the LR of TSHR and FSHR have anything in common. since this may give an explanation why the authors find that TSH makes contact with LR based on MD simulation but based on the FSH crystal data, no/weaker interaction exists.

[Editors’ note: further revisions were suggested prior to acceptance, as described below.]

Thank you for resubmitting your work entitled "Computational model of the full-length TSH receptor" for further consideration by *eLife*. Your revised article has been evaluated by previous reviewers, Carlos Isales (Senior Editor), and a Reviewing Editor.

The manuscript has been improved, but some remaining issues, as pointed out by Reviewer #2 need to be addressed, as outlined below:

1. While the authors did address many of my comments, I still think that the fact LR is disordered does not provide obvious mechanistic insights, and the simulations with the bound ligand are too preliminary to make solid conclusions. This was my #1 criticism, and the authors did not address it in their response; I didn't see it addressed in the manuscript either. To me, this is a fundamental oversight because readers will want to know the significance of these simulations. In my opinion, this significance is not spelled out clearly.

---

## [Author Response]

Reviewer #1 (Recommendations for the authors):Major weaknesses:– Page 29 line 508: The authors claim that this is the first model of full-length TSHR, but citation 39 appears to be a full-length TSHR homology model (with noted differences in LR conformation). This manuscript does appear to be the first simulation of full-length TSHR, however.

It is true that in the literature there are partial models of the TSHR based on homology of the linker (hinge) region but with an inability to model the 50 amino acid insert unique to the TSHR. We have made this point in the revised Discussion

The manuscript would benefit from stating more clearly what is in this model that previous models are lacking and why that is significant.

A statement has been added emphasizing that all previously published models are incomplete homology-based models whose coordinates are unpublished. We present here the first complete, physics-based model of the TSHR, especially the linker region, which provides details of the ectodomain conformations and orientation as well as a conformational ensemble generated by molecular dynamics. The significance of the results lies in our ability to obtain a structure of the LR using AF2 from its sequence, which allowed for the full length MD simulation model.

– Page 7 line 88: This paragraph describes the choice of orientation for the starting model. This model was selected first by rotating the LRD-LR around the z-axis and selecting an orientation that minimizes the volume of the box necessary for running the simulation and maximizes the number of contacts between LR and TMD (which the authors acknowledge as inexact). While this choice of starting model is computationally efficient, it is not sufficiently rigorous. The initial model appears to contain the interdomain hydrogen bonds that the authors track in Figure 7, and it is unclear that these bonds would be found and maintained if the starting conformation of the model were different. It is also unclear whether the TSHR-TSH interactions reported in Figure 10 would hold with a different starting conformation. To address this concern, the paper would benefit from another set of calculations where this arbitrary orientation is changed to be outside of the 45{degree sign} window that is presented to validate that these results are not dependent on initial orientation. It may also be valuable to report the orientation that AF2 produces (see next comment).

We agree that the heuristic process we used to arrive at the starting model lacks the simplicity and clarity of a simple optimization task. However, every optimization approach is dependent on the choice of the objective function, so there is always an arbitrariness in the choice of that objective function. On the other hand, the fact that during 1000ns the relative orientation/position of the LR and the TMD remained essentially the same in spite of the large conformational fluctuations of the LR and the movement of the LRD with respect to the rest of the TSHR indicates that whatever bias the initial model may have been affected by its effect was eliminated as the LR conformations developed. Further confirmation of the overall correctness of our initial model, noted in the Discussion section, is the fact that the starting model allowed the dimerization of the TSHR in such a way that the TSH binding surface remained free after dimerization. This, in our opinion made it unnecessary to pursue different starting orientations.

Note, also, that the interdomain hydrogen bonds (red lines in Figure 7) are bonds between the LR and the LRD, not between the LR and the TMD and they were present in the Alphafold2 structure. We added a note about this in the description of Figure 7.

– The authors used AF2 to determine the structure of the whole protein except for the Transmembrane domain (TMD), for which they had generated a model previously. How similar was the AF2 model of the whole protein to their model of two AF2 domains with the TMD? What does the TRIO model of the TMD give that an AF2 model of the TMD doesn't? AF2 has been shown to produce reasonable models of membrane bound proteins (Diego del Alamo, Davide Sala, Hassane S Mchaourab, Jens Meiler (2022) Sampling alternative conformational states of transporters and receptors with AlphaFold2 eLife 11:e75751); it would be an important addition if the authors could describe how/why AF2 cannot be used for this entire structure.

The AF2 program gave a separate model for the transmembrane domain and did not provide a fully assembled structure. We selected our own structure for the TMD as it was partially based on experimental data and already equilibrated in our membrane model. A comment has been added in the revision explaining the rationale behind our choice.

– It would be helpful if the authors could expand on the signaling mechanism mentioned in the last line of the discussion. This result and its mechanistic implications may be important in further studies (both computational and experimental) but it is barely discussed here.

We agree that this is an important question. In the Discussion we have added a hypothesis about the potential role of the disulfide bond(s) in TSHR signal transduction.

– Page 8 line 106: MD parameters necessary to run the six-step equilibration protocol and production run should be reported.

A reference to the protocol has been added.

– Page 10 line 156: This paragraph notes that many atoms clash and that the model requires further development. However, there is no mention of what was done to further develop the model. Any steps that were taken to further develop should be reported.

We have replaced the ambiguous statement “so the model required further development” by a statement to the effect that the MD simulation was expected to resolve these clashes and result in a superior model – which it did.

– Page 7 line 92: The authors should provide a rigorous definition of "mutually proximal". A distance threshold would be sufficient.

An explicit definition of “mutually proximal” has been added. The salient feature of this concept is that it does not involve a distance threshold.

– Page 9 line 135: the X-Y distance is below what threshold for hydrogen bonds?

Distance thresholds have been added.

Reviewer #2 (Recommendations for the authors):1. The authors should put more effort to provide clear, properly labeled and good quality figures.

All figures have been replaced with higher resolution versions

a. There is no labeling of figures like N-, C- terminal of the protein in Figure 2.

N and C labels added.

b. In Figure 4a, its very difficult to read the color codes below the figure.

All figures have been replaced with higher resolution. In addition, the color codes have been enlarged.

c. For fig6, I can hardly read the labels on y-axis.

All figures have been replaced with higher resolution

d. There is typo in line 364. it should be 284 and not 384.

Corrected.

2. In Figure 1, the authors showed the enlarged LR showing various epitopes. What is the purpose of this?

We removed the enlargement with the epitopes and the reference to it in the text.

3. The authors have used the pdb code 4AY9 for the structural analysis. They should also use sequence based analysis and find out whether the LR of TSHR and FSHR have anything in common. since this may give an explanation why the authors find that TSH makes contact with LR based on MD simulation but based on the FSH crystal data, no/weaker interaction exists.

This is an interesting suggestion to do a comparative sequence analysis on the LR between TSHR and FSHR. We found that there is only 39.3% sequence identity between the LR of TSHR and FSHR and added this information to the Discussion. Furthermore, we now note that the LR of FSHR lacks the cleaved 50aa region which is unique to the TSHR and thus misses the regions where our simulations show most of the TSHR-LR contacts.

[Editors’ note: further revisions were suggested prior to acceptance, as described below.]

1. While the authors did address many of my comments, I still think that the fact LR is disordered does not provide obvious mechanistic insights, and the simulations with the bound ligand are too preliminary to make solid conclusions. This was my #1 criticism, and the authors did not address it in their response; I didn't see it addressed in the manuscript either. To me, this is a fundamental oversight because readers will want to know the significance of these simulations. In my opinion, this significance is not spelled out clearly.

We added two paragraphs to the discussion summarizing our interpretation of the result of these calculations.